# Cryotherapy as a Surgical De-Escalation Strategy in Breast Cancer: Techniques, Complications, and Oncological Outcomes

**DOI:** 10.3390/biomedicines13122987

**Published:** 2025-12-05

**Authors:** Kai Lin Lee, Ashita Ashish Sule, Hao Xing Lai, Qin Xiang Ng, Serene Si Ning Goh

**Affiliations:** 1NUS Yong Loo Lin School of Medicine, National University of Singapore, Singapore 117597, Singapore; lee.kai.lin@u.nus.edu (K.L.L.);; 2Saw Swee Hock School of Public Health, National University Health System, Singapore 117549, Singapore; 3Department of General Surgery, National University Hospital, Singapore 119228, Singapore

**Keywords:** cryotherapy, breast cancer, ablation, recurrence, survival, de-escalation, minimally invasive therapy

## Abstract

**Background**: Early breast cancer outcomes have improved substantially, yet surgery may carry physical and psychosocial costs. Cryotherapy has gained attention as a minimally invasive alternative to surgery for select patients with breast cancer: particularly, those with small, unifocal, hormone receptor-positive tumors. Given rapidly expanding but heterogeneous reports, this state-of-the-art review therefore aims to synthesize information on how breast cryotherapy is performed, for whom it is most suitable, what outcomes to expect, and where evidence is still immature. **Methods**: We queried MEDLINE (via PubMed), Embase (via Ovid), and the Cochrane Library up to January 2025, using terms related to “breast neoplasms,” “cryotherapy,” and “cryoablation.” Eligible studies included clinical trials, cohort studies, and case series reporting outcomes of cryotherapy in breast cancer. Data were extracted on patient characteristics, procedural parameters, recurrence, survival, and complications. The risk of bias was assessed using the MINORS tool, and certainty of evidence was appraised with the GRADE framework. **Results**: A total of thirty one studies (comprising 1357 patients) formed the evidence corpus summarized here. Most involved early-stage, hormone receptor-positive breast cancers ≤ 2 cm treated with percutaneous cryoablation. Local recurrence, defined as any ipsilateral breast tumor recurrence confirmed radiologically or histologically, ranged from 0 to 68.8%, with smaller, unifocal tumors achieving the best control. Overall survival exceeded 80% in early-stage disease, while complications were generally minor, including bruising, hematoma, and skin erythema. Patient satisfaction was high, with favorable cosmetic outcomes reported in limited studies. However, the follow-up duration ranged from 1 month to 10 years (with nearly half < 1 year), and protocols varied substantially across studies. In summary, breast cryotherapy appears safe and can achieve encouraging local control and cosmetic results in carefully selected early-stage cases. Its role in aggressive subtypes, larger or multifocal disease, and as part of multimodal regimens requires further study. **Conclusions**: Standardized protocols, imaging/reporting conventions, and longer follow-up with patient-reported outcomes are needed to advance the field and further define where cryotherapy can appropriately de-escalate surgery.

## 1. Introduction

Today, breast cancer is the most common malignancy among women worldwide, with over 2.3 million new cases diagnosed annually [1]. Advances in screening and treatment have significantly improved outcomes, with five-year relative survival exceeding 90% for early-stage disease in high-income countries [2]. Current standard care typically involves curative resection, followed by adjuvant therapies. However, these approaches are not without consequences; they are associated with physical and psychological burdens, such as surgical morbidity, altered body image, and treatment-related complications [3]. These challenges have driven a growing interest in minimally invasive alternatives that offer oncological safety while preserving quality of life.

In response, cryotherapy has emerged as a potential surgical alternative for patients with small, solid breast tumors [4]. Preliminary studies have reported promising results in terms of local tumor control and patient satisfaction; it induces cell death through extreme cold, causing intracellular ice formation, vascular stasis, and apoptosis, while sparing surrounding healthy tissue [5]. Its precise, localized action allows for tumor ablation with favorable cosmetic outcomes, making it particularly suitable for patients who are poor surgical candidates [6].

However, significant heterogeneity in cryotherapy devices, treatment protocols, and patient populations has made it difficult to draw definitive conclusions regarding its long-term efficacy and safety. While the majority of early-phase clinical studies have focused on indolent tumors such as small, hormone receptor-positive lesions, emerging preclinical and translational research has begun to explore its use in more aggressive subtypes, including triple-negative and HER2-positive breast cancers. In particular, cryotherapy’s ability to induce immunogenic cell death and stimulate systemic anti-tumor immune responses has led to an increasing interest in its potential to elicit an abscopal effect, particularly in immune-responsive tumors such as triple-negative breast cancer [7,8,9,10].

This state-of-the-art review therefore aims to summarize the current evidence of cryotherapy for breast cancer, and evaluate the technical approaches, oncological outcomes, complications, and patient-centered metrics. The objective is to guide clinicians in understanding the role of cryotherapy in breast cancer management and to provide evidence-based insights for future research and standardization.

## 2. Methods

To anchor our synthesis, we referenced the PRISMA guidelines [11], and the review protocol was registered in PROSPERO a priori (registration number CRD42024579428).

We queried the MEDLINE (via PubMed), Embase (via Ovid), and Cochrane Library databases, using combinations of keywords including “breast neoplasms” and “cryotherapy”. The full search strategy is detailed in Appendix A. The screening and selection process was performed independently by two reviewers (K.L.L. and A.S.S.), who assessed titles and abstracts for relevance after removing duplicates. Any disagreements during the screening process were resolved through discussion, and unresolved cases were adjudicated by a third, senior reviewer (S.S.N.G).

References were managed using Rayyan version 1.5.0 (https://www.rayyan.ai accessed on 17 August 2024) [12] to streamline the screening and sieving process. Eligible studies were published in English up to January 2025, including randomized controlled trials, cohort studies, case–control studies, and cross-sectional studies evaluating cryotherapy with adjunct systemic interventions in breast cancer patients. Primary outcomes included recurrence rates and overall survival. Secondary outcomes included incomplete tumor necrosis and cosmetic outcomes. Studies that were limited to procedural feasibility without outcome data were excluded.

Two reviewers (K.L.L. and A.S.S.) then independently extracted data from the included studies, using a standardized data extraction form that was developed a priori and pilot-tested on a random sample of three studies. Extracted variables included study characteristics (author, year, country, design, sample size), patient demographics (age, menopausal status), tumor features (histological type, size, hormone receptor and HER2 status, focality), cryotherapy details (device type, number of probes, freeze–thaw cycles, iceball margin), and outcomes. To minimize potential double-counting, we cross-checked study authorship, centers, enrolment periods, and cryotherapy devices; no clear duplicate cohorts were identified, but multi-center registry-based studies were indicated in the results table (Table 1) where overlap could not be definitively excluded.

For this review, primary outcomes of interest were local recurrence, distant recurrence, residual disease, and overall survival. Secondary outcomes included procedural complications, post-treatment imaging findings, need for adjuvant therapy, and patient satisfaction metrics. Where available, we extracted the follow-up duration and imaging modalities used for treatment response evaluation (e.g., MRI, PET-CT, contrast-enhanced mammography). We categorized iceball margins into three groups (<0.5 cm, 0.5–1.49 cm, ≥1.5 cm) and classified the complications by severity and frequency. For recurrence and survival outcomes, we noted the timing of recurrence, any reintervention performed (e.g., re-cryoablation or surgery), and the type of systemic therapy administered post-cryoablation. Any disagreements in data extraction were resolved by a discussion with a third reviewer (S.S.N.G).

Due to heterogeneity in the study design, patient populations, and outcome reporting, meta-analysis was not feasible. Instead, we conducted a narrative synthesis and stratified the results by tumor subtype, procedural technique, and treatment setting (e.g., early-stage vs. metastatic). Patterns were identified in recurrence rates based on technical adequacy (e.g., iceball margins, probe number), and residual disease was evaluated in relation to tumor size and histological subtype.

Two reviewers (A.S.S. and H.X.L.) independently assessed the risk of bias, using the Methodological Index for Non-Randomized Studies (MINORS) [13]. Certainty of evidence was evaluated using the GRADE (Grading of Recommendations Assessment, Development, and Evaluation) framework, considering the risk of bias, inconsistency, indirectness, imprecision, and publication bias, with ratings categorized as high, moderate, low, or very low [14], based on team consensus.
biomedicines-13-02987-t001_Table 1Table 1Summary of study and patient characteristics across reviewed studies.Study (Year)CountryStudy Design (*n*)Key Inclusion CriteriaTumor Size (cm)InterventionCryotherapy ProtocolFollow-Up (mo)Cazzato (2015) [15]FranceProspective Cohort (*n* = 23)Biopsy-proven BC; ≥0.5 cm tumor–skin distance≤3.0Percutaneous cryoablationIceSphere/IceRod; 1–2 sessionsNRComen (2024) [10]USAProspective Cohort (*n* = 5)Early-stage IDC (T1/T2), no mets≥1.5Cryo + ipilimumab/nivolumabIce Pearl/Ice Force; 1 session~47Gajda (2014) [16]GermanyCohort (*n* = 53)Biopsy-proven BC≤1.5 in 32 pts; >1.5 in 20 ptsPercutaneous cryoablationNot reported; surgery after cryo1–35 days *Habrawi (2021) [17]USAProspective Cohort (*n* = 12)Age ≥ 50, low-risk unifocal IDC~1.0Percutaneous cryoablationVisica^®^ 2; single sessionNRKawamoto (2024) [18]JapanProspective Cohort (*n* = 18)Early-stage IDC~1.0Percutaneous cryoablationProSense; single session~Not reportedKhan (2023) [19]USACohort (*n* = 32)Age ≥ 50, unifocal IDC, ≤T1/T20.87 ± 0.35Percutaneous cryoablationVisica 2/ProSense; single session36Kwong (2023) [20]Hong KongProspective Cohort (*n* = 15)Solitary T1 BC;≥0.5 cm to skin1.3 by USPercutaneous cryoablationProSense; single sessionNRRzaca (2013) [21]PolandCase Series (*n* = 6)Paget’s disease of nippleNRPercutaneous cryoablationKriopol K26-M1; 1–2 sessions~94Pusztaszeri (2007) [22]SwitzerlandProspective Cohort (*n* = 11)US-visible invasive BC<2.0Percutaneous cryoablationCRYO-HIT; single session4–5 wks *Poplack (2015) [23]Int’lProspective Cohort (*n* = 20)IDC ≤ 1.5 cm1.0 (mean)Percutaneous cryoablationVisica/Visica 2; single session~1Pfleiderer (2002) [24]GermanyCohort (*n* = 15)US-detectable IDC or ILC; ≥1 cm from skin2.1 ± 0.78Percutaneous cryoablationCryoHit; single session1–5 days *Manenti (2013) [25]ItalyRetrospective Cohort (*n* = 80)Subclinical IDC (≤2.0 cm)~1.6Percutaneous cryoablationIceRod; single session18McArthur (2016) [26]USAProspective Cohort (*n* = 19)Early-stage BC, T ≥ 1.5 m≥1.5Cryo ± ipilimumabIceRod/IceSeed; single session~31Beji (2017) [27]FranceProspective Cohort (*n* = 17)Metastatic IDC/ILC0.5–4.5Percutaneous cryoablationIceSeed/IceRod; single session22Kinoshita (2017) [28]JapanCase Series (*n* = 4)Small BC, “Luminal A”~1.13Percutaneous cryoablationCryoHit; single session6Pfleiderer (2005) [29]GermanyProspective Cohort (*n* = 30)Stage T1 BC ≤ 1.5 cm0.5–1.5Percutaneous cryoablation0.33 cm Galil cryoprobe; single sessionNRVogl (2024) [30]GermanyRetrospective Cohort (*n* = 45)Unresectable BC ≤ 3 lesions1.6 ± 0.7Percutaneous cryoablationIceCure ProSense; mostly single session24Navarro (2023) [31]SpainProspective Cohort (*n* = 20)Low-risk, unifocal IDC~0.9Percutaneous cryoablationICEfx Galil; single session25 days *Oueidat (2024) [32]USARetrospective Multi-Center (*n* = 112)BC ineligible for trials~1.0 (0.7–1.8)Percutaneous cryoablationVarious devices; single session24Adachi (2020) [33]JapanRetrospective Cohort (*n* = 193)Early BC (IDC/DCIS)0.9 (0.25–1.5)Percutaneous cryoablationVisica I/IceSense3; single session12Simmons (2016) [34]USAProspective Cohort (*n* = 86)Unifocal IDC ≤ 25% in situ0.1–2.0Percutaneous cryoablationVisica 2; sessions NR~1Niu (2013) [35]ChinaProspective Cohort (*n* = 120)Metastatic BC>5.0 cm used 3–4 probesCryo ± immuno/chemoEndocare argon-based; single vs. multiple120Fine (2024) [36]USAProspective Cohort (*n* = 194)Early-stage, US-visible BC0.28–1.4Percutaneous cryoablationProSense; freeze cycles tailored to size54Pusceddu (2017) [37]ItalyProspective Cohort (*n* = 35)BC with metastatic disease3.0 ± 1.4Percutaneous cryoablationGalil (SeedNet); 1–2 sessions46Manenti (2011) [38]USAProspective Cohort (*n* = 15)DCIS ≤ 1.0 cm0.4–1.2Percutaneous cryoablationIceRod, single session6Littrup (2009) [39]USAProspective Cohort (*n* = 11)Biopsy-proven BCIS1.7 ± 1.2Multi-probe cryoablationEndocare cryoprobes; 1–2 sessions18Kawamoto (2021) [40]JapanProspective Cohort (*n* = 8)IDC ≤ 15 mm, HER2-, Ki-67 ≤ 20%~1.0US-guided cryoablation + RT and AIProSense 10 G; single session28Machida (2018) [41]JapanRetrospective Cohort (*n* = 54)DCIS or IDC ≤ 1.5 cm~0.9Cryoablation onlyVisica I or IceSense3; 1–2 sessions41Liang (2017) [42]ChinaProspective Cohort (*n* = 16)HER2+ recurrent BC, KPS ≥ 60NRCryo + NK cells + HerceptinEndocare argon-helium; single session9–12Sabel (2004) [43]MultinationalProspective Cohort (*n* = 29)Solitary BC ≤ 2.0 cm~1.2 ± 0.5CryoablationVisica system; single sessionNRNavarro (2024) [44]Spain Prospective Cohort (*n* = 59) Age ≥ 18, suitable for BCS, no requirement for primary systemic therapy, IDC ≤ 2.0 cm, ER+/HER2-, negative axillary status 1.01 ± 0.36CryoablationICEfx Galil; single session22 days *Abbreviations: USA= United States of America, BC = breast cancer; IDC = invasive ductal carcinoma; ILC = invasive lobular carcinoma; DCIS = ductal carcinoma in situ; BCIS = breast carcinoma in situ; RT = radiotherapy; AI = aromatase inhibitor; KPS = Karnofsky performance status; NR = not reported; T1/T2 = tumor stage; US = ultrasound; BCS = breast conserving surgery. * Follow-up was reported in days or weeks for these studies, rather than in months; Wks = weeks, mo = month.


## 3. Results

As illustrated in Figure 1, a total of 276 records were screened and assessed for eligibility. After applying the predefined inclusion and exclusion criteria, 31 studies [10,15,16,17,18,19,20,21,22,23,24,25,26,27,28,30,31,32,33,34,35,36,37,38,39,40,41,43,44] with a total of 1357 adult patients were selected for inclusion.

### 3.1. Patient Selection and Tumor Characteristics

Across cohorts, most candidates had small (≤2–3 cm), unifocal, hormone receptor–positive invasive ductal carcinomas; DCIS and selected metastatic/palliative contexts were also reported. Multifocal disease, lobular histology, higher grade, and tumors >2 cm were repeatedly associated with residual disease or higher recurrence, underscoring the importance of careful selection [20,33,34,35]. Study characteristics and patient characteristics across the studies included in this review are summarized in Table 1.

Notably, 26 studies used percutaneous cryotherapy alone, while 5 combined it with other therapies. The age range was 53–85 years. Comparators included ipilimumab [10] and radiofrequency ablation [25]. Some examined cryotherapy combined with immunotherapy or chemotherapy [26,35].

With regard to breast cancer characteristics, 29 studies focused on primary non metastatic breast cancer and 2 were on metastatic cancer, primarily for local palliation or debulking. Tumors were <3 cm in 24 of studies, with the largest being reported at 4.5 cm [27]. Invasive ductal carcinoma (80%) was most common, followed by invasive lobular carcinoma (30%) and ductal carcinoma in situ (30%). Hormone receptor-positive tumors dominated, with only two studies including triple-negative breast cancer.

As for cryotherapy device characteristics, liquid nitrogen or argon gas-based devices were used, with some studies using multiple systems. Notably, seven (23%) studies used the ProSense ablation system (IceCure Medical Ltd., Yokneam, Israel), five (17%) used Visica 2 (Sanarus Technologies, Inc., Pleasanton, CA, USA), four (13%) used IceRod (Galil Medical, Yokneam, Israel), and four (13%) used Visica 1 (Sanarus Medical, CA, USA). A summary of all systems used is found in Figure 2 and Table 1.

### 3.2. Technique and Protocol for Cryotherapy

In terms of the number of cryotherapy sessions, 23 (74%) studies had single cryotherapy sessions. Of the remaining eight (27%) studies, five (17%) treated local recurrences with a second cryotherapy session, one (3%) treated a new breast lesion discrete from the original ablated site, one (3%) treated ipsilateral recurrence with both cryoablation and total mastectomy, and one (3%) compared single versus multiple cryotherapy sessions.

As for the number of probes, of the 22 reported studies, 12 (55%) used a single probe, and 10 (45%) used multiple probes for larger tumors. Pusztaszeri et al. [22] used one probe for tumor sizes <1.2 cm and two needles for tumors ≥1.2 cm. In two (7%) studies, Niu et al. [27] and Liang et al. [42], three to four cryoprobes were used for larger masses, defined as >5.0 cm. In terms of iceball diameters, 11 studies (27%) achieved an iceball margin of 1.0–1.49 cm, 6 studies achieved 0.5–0.99 cm, and 3 studies achieved ≥1.5 cm.

Regarding the freeze–thaw protocol, 2 freeze–thaw cycles were the most common (67%), while 3 cycles were used in 10%. The duration of each freeze–thaw cycle ranged from 6 to 10 min.

Notably, to prevent skin burns, more than half (57%) of studies utilized saline injections into subcutaneous tissue, with warm saline being the most common approach. Other methods included hemocoagulase [42] and fibrin glue [35].

### 3.3. Imaging Follow-Up and Response Assessment

Regarding post-cryotherapy imaging, 14 studies used Magnetic Resonance Imaging (MRI) (47%), 11 studies used ultrasound (37%), and 3 studies used Computed Tomography (CT) scans and FDG-PET CT scans (10%). Some studies used a combination. The most common finding post-cryotherapy is the loss of intense enhancement [23,25,27,28,34,37,38], fat necrosis [17,25,33,38,40], and reduced lesion size [18,26,41,42].

To assess the response, 20 (67%) studies assessed the response based on imaging. Based on breast-enhanced spiral CT using RECIST criteria, Liang et al. [42] reported complete regression in three (6%) cases and partial regression in eight (15%) cases, while three studies used histology. Simmons et al. [34] reported successful cryoablation in 66 (76%) cancers. Comen et al. [10] assessed the response based on T-cell activation markers and serum Th1 cytokines. Predictors of an incomplete response included multifocal disease, large tumor size, and a molecular subtype of the tumor and grade.

To assess residual disease, 12 (40%) studies evaluated the residual disease status with post-cryotherapy imaging and/or pathological analysis of tumorectomy specimens, with rates ranging from 0 to 69% [16,20,24]. Gajda et al. [16] reported 53.3% residual carcinoma, in which the tumor-free cases after cryotherapy were limited to those with cT1 stage tumors. Kwong et al. [20] found that tumor biology did not affect the center of the cryozone, but 47% of lumpectomy specimens had residual cancer at the periphery. Across studies, smaller and unifocal tumors (<1.5 cm) achieved higher rates of complete ablation and a lower recurrence than larger or multifocal lesions. Tumors > 2 cm were more likely to show residual disease, due to incomplete iceball coverage. Pusztaszeri et al. [22] and Gajda et al. [16] reported residual carcinoma mainly in larger lesions, while Fine et al. [36] observed durable local control with a 4% five-year recurrence rate in predominantly T1 cancers. Lesions that were multifocal, high-grade, or of lobular histology were more prone to incomplete response, underscoring the influence of both anatomical and biological factors on cryotherapy efficacy and the importance of careful patient selection.

As for surgery or adjuvant therapy post-cryotherapy, 12 studies included surgery post-cryotherapy, with 7 studies performing lumpectomy and mastectomy, and 5 with lumpectomy only. Three studies performed Sentinel Lymph Node Biopsies (SLNBs) post-cryotherapy, with SLNB rates ranging from 97 to 100% [25,28,34]. Two-thirds of the studies reported axillary lymph node dissection in 0–38% of cases. All patients with positive SLNB had axillary clearance, except one who subsequently received adjuvant therapy only.

Adjuvant therapy data were inconsistently reported. The exact number of patients receiving each therapy was not consistently reported in all studies and was only noted where available. Radiotherapy was reported in five studies, hormonal therapy in four, chemotherapy in one, and endocrine therapy in one. Kinoshita et al. [28] treated all four patients with endocrine and radiotherapy. In Kawamoto et al. [40], all eight patients received whole breast irradiation 50 Gy/25 fractions and endocrine therapy. In Machida et al. [41], all 54 patients received adjunctive radiotherapy 34–181 days after cryoablation to the ipsilateral breast.

### 3.4. Patient Satisfaction and Complications

The majority of the studies did not report patient satisfaction (*n* = 24, 80%). In the remaining six studies, the median follow-up ranged from 12 to 54 months. Kawamoto et al. [18] reported no nipple distortion, breast deformity and asymmetry with the five-point scale, the EuroQol Visual Analog Scale and the EuroQol-5 Dimensions-5 Level, while Kawamoto et al. [40] reported high satisfaction. Three studies reported good patient satisfaction [25,36,39]. Liang et al., 2017 used Karnofsky performance status (KPS) and found improved functional outcomes post-cryotherapy [42].

Pertaining to complications, 17/30 studies reported on complications, and of these, 4/17 studies had none. Kwong et al. [20] noted no frost injury, bleeding, and hematoma at 8 weeks. Commonly reported complications included hematomas (24%, *n* = 4), bruising (24%, *n* = 4), erythema (18%, *n* = 3), skin burns (12%, *n* = 2), edema (12%, *n* = 2), and rash (6%, *n* = 1).

### 3.5. Oncological Outcomes

Follow-up duration across studies ranged from 1 day [16,24] to 10 years [35]. Of the included cohorts, 13 studies (43%) had <1 year follow-up, 7 (23%) had <2 years, and 4 (13%) had <3 years, limiting the interpretation of long-term oncologic outcomes. Among the 28/30 studies evaluating cryotherapy for primary, early-stage breast cancer, local recurrence rates varied widely: nine studies (32%) reported 0% recurrence, five (18%) reported <10%, one (4%) reported <20%, three (11%) reported <30%, and one (4%) reported <40%. Notably, nine studies (32%) did not report local recurrence at all. Six early-stage studies reported managing recurrence with repeat cryotherapy [15,21,27,32,37,41]. Distant recurrence in this curative-intent group was reported in 14 studies, with 11 showing 0%, 2 reporting <10%, and 1 reporting <20%. Vogl et al. [30] reported the highest rate of intramammary distant recurrence (13%, *n* = 6). Overall survival was inconsistently reported: only 13/28 (46%) of early-stage studies documented overall survival outcomes, with nine studies reporting 100% overall survival and four reporting overall survival between 80 and 92% [30,36,37].

In contrast, outcomes from metastatic or palliative-intent cohorts were reported separately and cannot be interpreted alongside curative-intent data. Liang et al. [42], examining HER2-overexpressing recurrent disease, demonstrated longer progression-free survival in the combination cryotherapy–NK cell therapy group compared with cryotherapy alone (12 vs. 9 months), which is consistent with the treatment administered for disease control rather than a definitive local cure.

### 3.6. Risk of Bias and Certainty of Evidence

The risk of bias was evaluated using the MINORS and most studies included were of a moderate methodological quality, with the total scores ranging from 7 to 21 (detailed in Appendix A). The majority of studies (*n* = 26) were non-comparative, reflecting early-phase or feasibility designs, while only a few comparative studies achieved higher scores (≥16). Common limitations included lack of prospective sample size calculation, short or variable follow-up periods, and an absence of a blind endpoint assessment. Nevertheless, most studies clearly stated their aims, used consecutive patient inclusion, and applied endpoints that were appropriate to their objectives. Only a small subset of recent prospective studies (e.g., Comen et al. [10] and Liang et al. [42]) achieved a good methodological quality, indicating that evidence supporting cryotherapy remains promising but methodologically heterogeneous.

Using the GRADE framework, the certainty of evidence across key outcomes ranged from low to high (Table 2). Evidence for OS was rated high in non-metastatic cases, supported by consistent findings across multiple prospective cohorts, showing survival exceeding 80%. In contrast, evidence for local recurrence and residual disease was graded as moderate, primarily due to substantial heterogeneity in tumor characteristics (23/30 studies restricted to ≤2–3 cm tumors; some included lesions up to 4.5 cm), procedural techniques (iceball margins ranged from 0.5 to 1.5 cm; freeze–thaw cycles ranged from two cycles in 67% of studies to three cycles in 10%; probe number differing from one probe in 55% of studies to multiple in the rest), and follow-up duration. Studies consistently demonstrated lower recurrence in small, unifocal, hormone receptor-positive tumors, but reporting variability and limited long-term data reduced confidence in pooled estimates. Patient satisfaction evidence was low-certainty, given that only a few studies assessed cosmetic or quality-of-life outcomes systematically. Overall, while cryotherapy appears to offer favorable outcomes in select early-stage breast cancers, the evidence base remains methodologically diverse, warranting more standardized protocols, longer follow-up, and patient-centered outcome reporting to strengthen future GRADE confidence ratings.

## 4. Discussion

### 4.1. Practical Implications (Clinician Takeaways)

The findings of this review indicate that cryotherapy shows the greatest promise in early-stage, hormone receptor-positive tumors. Preliminary evidence in late-stage or metastatic settings suggests potential palliative benefits, particularly in pain control and local tumor burden reduction, but requires more robust data to verify any survival advantage [18,19]. Triple-negative breast cancers (TNBC) and HER2-positive tumors present greater challenges, due to their aggressive nature and poorer prognosis [45,46]. TNBC, characterized by the absence of ER, PR, and HER2 receptors, lacks targeted hormonal therapies, making its management more complex [45,46]. This necessitates more aggressive treatment modalities, such as chemotherapy or immunotherapy, which may limit the exploration of cryotherapy as a viable option. HER2-positive tumors often require targeted therapies like trastuzumab, which complicate the integration of cryotherapy into standard treatment protocols [31,47,48]. The prioritization of hormone receptor-positive tumors in research highlights a need for further studies on cryotherapy’s role in aggressive subtypes.

Interestingly, there is marked variability amongst cryotherapy protocols, including device types (argon vs. liquid nitrogen), probe sizes, freeze–thaw cycles, and margin definitions (0.5–1.5 cm). This variability complicates direct cross-study comparisons and highlights the need for standardized cryotherapy protocols, which would likely facilitate better reproducibility and a clearer evaluation of the outcomes. The ProSense system, with two cycles per session, is most commonly used [18,19,36]. Some had fixed timings for each freeze and thaw phase [15,22,23,25,26,27,28,33,41], while others had more flexible timings with a primary intent of achieving a minimum iceball margin [15,36,38]. Some studies report additional cryotherapy sessions due to recurrence, raising questions about long-term efficacy [15,21,23,27]. Prevention methods for skin burns, such as saline injections or warmed saline bags, vary in effectiveness. Injected saline is sometimes reabsorbed too quickly to form an adequate barrier, while warmed saline bags appear to be more effective for preventing injury [15,26,31,32,37]. The use of multiple probes in larger tumors to achieve a more comprehensive and uniform destruction of tumor tissue suggests a shift toward personalized treatment [25]. Larger tumors present a unique challenge in achieving adequate ablation margins, as the size may exceed the freezing capabilities of a single probe, potentially leading to incomplete necrosis and subsequent tumor recurrence. To address these inconsistencies, we propose a provisional technical framework derived from the most frequently reported parameters across the studies included. At minimum, protocols should incorporate: (1) a double freeze–thaw cycle; (2) a target iceball margin ≥ 10 mm beyond the tumor; (3) explicit documentation of probe count and configuration, with guidance for multi-probe placement in larger tumors; (4) standardized reporting of freeze and thaw timing, whether fixed or margin-driven; and (5) mandatory skin-protection measures, with warmed saline bags being recommended as the preferred technique. We further suggest a reporting template that includes detailed documentation of tumor size, distance to skin, intended margins, probe type and number, device system, real-time monitoring modality, and standardized follow-up imaging windows.

Accurate assessment of the residual tumor after cryoablation is essential for determining treatment success and guiding subsequent interventions. Although MRI remains the preferred modality because of its superior soft-tissue contrast and sensitivity for detecting residual enhancement, its performance is not flawless in the post-cryoablation setting. False-negative MRI examinations can occur, particularly when early scans are confounded by inflammatory changes or evolving fat necrosis, both of which may obscure subtle residual disease [49]. Fat necrosis can also mimic either residual enhancement or obscure the ablation cavity’s expected involution pattern, making interpretation challenging, even for experienced breast radiologists. PET-CT may offer additional value when metabolic activity is needed to differentiate a viable tumor from a post-treatment change, but it is expensive and generally less spatially precise for a small-volume breast disease [50]. The financial burden of serial high-cost imaging further complicates routine long-term monitoring for both patients and healthcare systems. This underscores the potential utility of cost-effective alternatives such as contrast-enhanced mammography (CEM), which may offer sufficient sensitivity for detecting suspicious enhancement while reducing cumulative imaging costs [8]. However, no study has yet provided a comprehensive cost-effectiveness evaluation comparing MRI, PET-CT, CEM, and multimodal strategies, representing a key gap in the current evidence.

Synthesizing these strengths and limitations suggests that post-cryoablation surveillance should be intentionally staged and risk adapted. Early MRI (<3 months) risks false-negatives due to incomplete ablation zone maturation; therefore, a 3 month baseline MRI or CEM, followed by interval imaging at 9–18 months, depending on the BI-RADS category, may strike a balance between diagnostic accuracy and resource stewardship [51]. Escalation to advanced imaging or biopsy should be triggered by the features that are inconsistent with expected cavity involution, such as new or enlarging enhancement, asymmetrical wall thickening, or metabolic activity that is not attributable to fat necrosis.

In terms of complications, hematomas and bruising were the most common, though up to 80% of patients in some studies reported no adverse events, raising questions about reporting biases and the importance of rigorous follow-up to capture adverse events comprehensively [15,17,25,35]. Patients who have bleeding diathesis should be counseled and optimized appropriately prior to cryotherapy, to pre-empt them of complications such as hematomas and bruising. The documented adverse events, while concerning, appear manageable, emphasizing the need for proper patient selection and monitoring. Patient satisfaction was linked to breast cosmesis and nipple preservation [18,40], though the persistence of palpable lumps post-treatment remains underreported.

Despite the advantages of cryotherapy, such as good cosmesis, minimal downtime, and outpatient feasibility, it should be viewed as complementary to, rather than a replacement for, established standards of care. Breast-conserving surgery with radiotherapy remains the primary modality for durable long-term local control, while endocrine therapy alone may be appropriate only for highly selected low-risk luminal A patients. Table 3 outlines a comparative overview of these de-escalation approaches to guide clinical decision-making in populations that are similar to those in ICE3.

### 4.2. Strengths, Limitations and Future Directions

This review provides the first comprehensive synthesis of cryotherapy outcomes across a substantial number of studies spanning diverse geographical settings and methodological designs, thereby strengthening the breadth and depth of available evidence for its use in breast cancer treatment. The application of the GRADE framework further enhances transparency by systematically evaluating the certainty of the evidence. Additionally, the identification of potential predictive factors for successful cryotherapy offers clinically relevant insights that may support patient selection and pre-treatment planning. Nonetheless, several limitations should be acknowledged. The predominance of studies involving Caucasian populations (*n* = 22), compared with a smaller representation of Asian cohorts (*n* = 8), may constrain the generalizability of findings across diverse patient groups. Nearly half of the included studies (*n* = 13) reported follow-up durations of less than one year, limiting the assessment of long-term local control and survival; future research would benefit from extended and standardized follow-up periods. Patient-reported outcomes were reported in only six studies, and the use of heterogeneous instruments (EQ-5D-5L, EuroQol VAS, 5-point cosmesis scales, KPS) contributed to variability in the outcome measurement. Although commonly evaluated domains included pain, appearance, and functional recovery, the field would benefit from the development and adoption of a standardized core patient-reported outcome set, to ensure consistency and comparability across future trials. Moreover, reporting of radiotherapy dose and adjuvant therapy details was inconsistent, limiting the interpretation of their roles as potential predictors of cryotherapy failure.

Further research should explore the impact of margins, radiotherapy regimens, and systemic therapies on long-term outcomes. Finally, beyond cryotherapy, other non-surgical modalities, such as whole-breast radiotherapy or endocrine therapy, have been proposed as de-escalation options for medically inoperable or biologically low-risk early breast cancer. Comparative evaluations incorporating cost-effectiveness and quality-of-life outcomes will be essential to delineate the optimal therapeutic strategy for this patient population.

## 5. Conclusions

In summary, cryotherapy represents a promising step toward therapeutic de-escalation in the management of select breast cancer populations, aligning with the broader shift toward personalized and less invasive care. The current evidence indicates that, in appropriately selected patients with small, unifocal, hormone receptor-positive tumors, cryoablation can achieve excellent local control, high patient satisfaction, and low complication rates. However, the evidence base remains fragmented and is predominantly limited to early-phase or non-comparative studies. To establish cryotherapy as a viable alternative or adjunct to surgical resection, future research should focus on standardizing procedural protocols, extending follow-up durations, and capturing long-term oncologic and patient-reported outcomes. Expanding the investigation into more aggressive subtypes and integrating cryotherapy within multimodal treatment frameworks will also be critical to defining its role in contemporary breast cancer care.

## Figures and Tables

**Figure 1 biomedicines-13-02987-f001:**
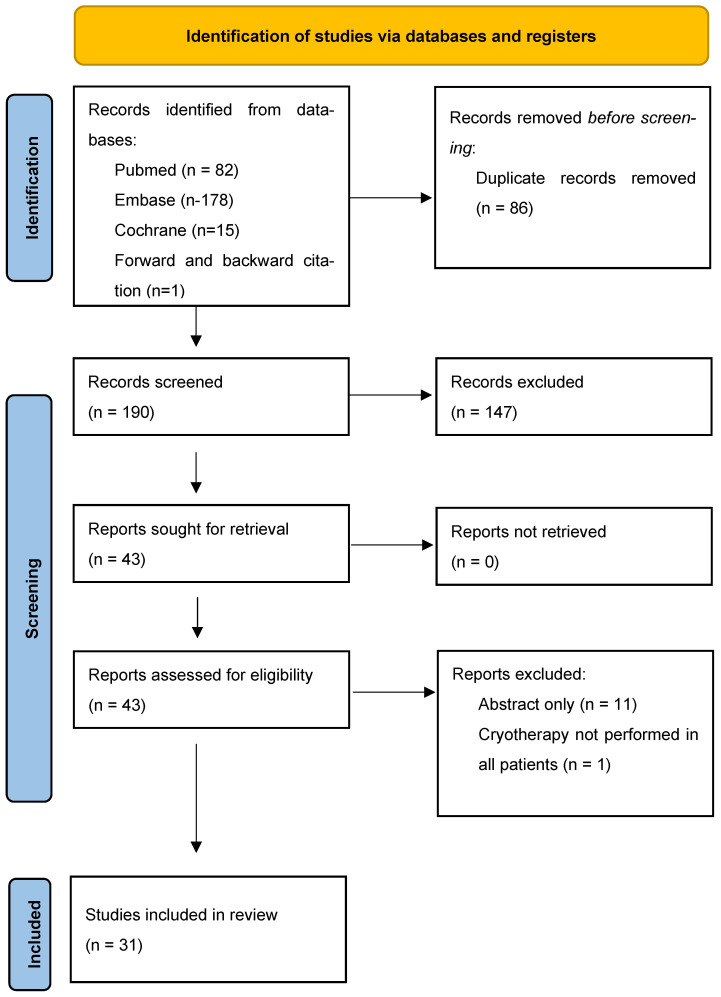
PRISMA flowchart showing the study selection process.

**Figure 2 biomedicines-13-02987-f002:**
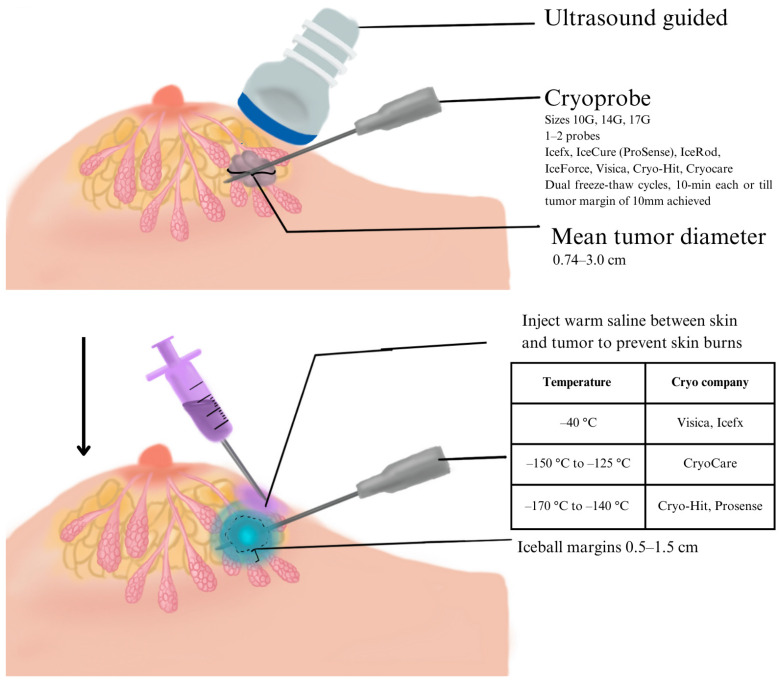
Schematic illustration of ultrasound-guided percutaneous breast cryoablation. The procedure involves real-time ultrasound guidance for accurate cryoprobe placement (10 G–17 G, typically one to two probes), targeting small, unifocal breast tumors (mean diameter 0.74–3.0 cm). Dual freeze–thaw cycles (approximately 10 min each) are applied until a 10 mm tumor margin is achieved. Warm saline is injected between the skin and the tumor to prevent thermal injury. The “iceball” margin typically extends 0.5–1.5 cm beyond the tumor boundary. Common cryotherapy systems include Visica, Icefx, IceCure (ProSense), IceRod, CryoCare, and Cryo-Hit, with operating temperatures ranging from −40 °C to −170 °C, depending on the device.

**Table 2 biomedicines-13-02987-t002:** Assessment of certainty of evidence of main outcomes as per GRADE.

Outcome	Certainty (GRADE) ^1^	Key Findings	Reasons for Downgrading or Upgrading
Local Recurrence	⊕⊕⊕O Moderate	Recurrence rates ranged from 0% to 68.8%; lower recurrence in small/unifocal tumors	High variability in outcomes; limited follow-up periods
Overall Survival	⊕⊕⊕⊕ High (non-MBC)	Survival > 80% for early-stage cases; metastatic survival extended with multiple cryotherapy sessions	Robust data for early-stage; limited studies for metastases
Residual Disease	⊕⊕⊕O Moderate	Higher residual disease with larger tumors or multifocal disease	Inconsistent reporting, heterogeneity in tumor characteristics
Patient Satisfaction	⊕⊕OO Low	High satisfaction with cosmesis reported in limited studies	Underreporting in 80% of studies

^1^ ⊕⊕⊕⊕ = High certainty, ⊕⊕⊕O = Moderate certainty, ⊕⊕OO = Low certainty.

**Table 3 biomedicines-13-02987-t003:** Comparative overview of de-escalation strategies for early breast cancer.

Domain	Cryotherapy	Breast-Conserving Surgery + Radiotherapy (BCS + RT)	Endocrine Therapy Alone (Selected Luminal A)
Local control (5–10 yr)	Emerging evidence; limited long-term data. ICE3: 4% 5-yr IBTR in T1 tumors; risk increases with >1.5–2 cm, multifocality, lobular histology.	Gold standard. IBTR ~2–5% at 5 yr with modern RT; most reliable long-term durability.	Weakest local control. Historical series: IBTR 10–20% +, often unacceptable except in frail medically inoperable patients.
Complications	Mostly minor: bruising, hematoma, fat necrosis; rare skin burns. No general anesthesia.	Surgical + RT toxicities: wound complications, seroma, fibrosis, skin changes, fatigue.	Minimal procedural complications; endocrine side-effects (arthralgia, hot flashes, mood changes).
Cosmesis	Excellent; minimal scarring; no volume loss.	Good but variable; may see contour changes, nipple displacement, fibrosis, depending on RT.	Excellent; no procedural changes.
Quality of life (QoL)	High satisfaction in reported studies, but PRO data limited.	Good long-term QoL; transient RT-related fatigue.	Depends on endocrine tolerance; some discontinue due to side effects.
Surveillance burden	Higher early imaging burden (MRI/US/CEM) to detect residual disease.	Standard annual mammography.	Standard annual mammography.
Suitability criteria	Best for the following: small (<1.5–2 cm), unifocal, HR+, IDC, US-visible tumors; patients desiring non-surgical options.	Broadly suitable for most early-stage breast cancers.	Only for very frail or medically inoperable low-risk luminal A patients who decline/are unfit for surgery.
Retreatment options	Possible repeat cryo; conversion to surgery feasible.	Re-excision or mastectomy if recurrence.	Surgery if disease progresses with endocrine therapy.
Logistical/cost considerations	Outpatient; lower procedural cost compared to conventional surgery; advanced imaging follow-up may increase cost.	Highest resource burden (procedural cost, theater utilization).	Lowest procedural cost; long-term endocrine therapy monitoring needed.

Abbreviations: IBTR = ipsilateral breast tumor recurrence; BCS = breast-conserving surgery; RT = radiotherapy; HR+ = hormone receptor-positive; IDC = invasive ductal carcinoma; US = ultrasound; CEM = contrast-enhanced mammography; QoL = quality of life; PRO = patient-reported outcome.

## Data Availability

No new data were created.

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
