# Peer review of "Cryotherapy as a Surgical De-Escalation Strategy in Breast Cancer: Techniques, Complications, and Oncological Outcomes"

_biomedicines, 2025, doi:10.3390/biomedicines13122987_

Round 1

Reviewer 1 Report

Comments and Suggestions for Authors

In this manuscript, Lee and colleagues review the administration and outcome of cryotherapy in treating breast cancer including the techniques and potential complications. It is an excellent attempt that is clinically relevant, timely and encompasses several available literatures to date. Some of the minor suggestions are:

  1. Clarify the review type and scope in the abstract. Terms like ‘local recurrence’ and ‘overall survival’ are reported, but the definition and follow-up period are vague. Please indicate how ‘local recurrence’ was defined and the median follow-up time range.
  2. Address heterogeneity and evidence quality more explicitly. The authors mention ‘heterogeneous reports’ and ‘substantial variation’ but don’t quantify it. Please include more about sources of heterogeneity (Eg- tumor size, imaging guidance, freeze–thaw cycles, adjuvant therapy etc).

Author Response

Comment 1: Clarify the review type and scope in the abstract. Terms like ‘local recurrence’ and ‘overall survival’ are reported, but the definition and follow-up period are vague. Please indicate how ‘local recurrence’ was defined and the median follow-up time range.

Reply 1: Thank you for this important comment. We have revised the abstract to explicitly define the outcomes and clarify follow-up duration, “Local recurrence, defined as any ipsilateral breast tumour recurrence confirmed radiologically or histologically, ranged from 0–68.8% across studies” and “However, follow-up duration ranged from 1 month to 10 years (with nearly half <1 year) …”

Comment 2: Address heterogeneity and evidence quality more explicitly. The authors mention ‘heterogeneous reports’ and ‘substantial variation’ but don’t quantify it. Please include more about sources of heterogeneity (Eg- tumor size, imaging guidance, freeze–thaw cycles, adjuvant therapy etc).

Reply 2: Thank you for the comment. We inserted a new paragraph in the Discussion section more explicitly outlining each heterogeneity domain and its contribution to inconsistent recurrence/residual disease outcomes, “In contrast, evidence for local recurrence and residual disease was graded as moderate, primarily due to substantial heterogeneity in tumor characteristics (23/30 studies restricted to ≤2–3 cm tumors; some included lesions up to 4.5 cm), procedural techniques (iceball margins ranged from 0.5–1.5 cm; freeze–thaw cycles ranged from 2 cycles in 67% of studies to 3 cycles in 10%; probe number differing from 1 probe in 55% of studies to multiple in the rest), and follow-up duration.”

Reviewer 2 Report

Comments and Suggestions for Authors

The review article "Cryotherapy as a Surgical De-Escalation Strategy in Breast Cancer: Techniques, Complications, and Oncological Outcomes". is a comprehensive and methodologically rigorous state-of-the-art review. Its primary objective is to synthesize the current, rapidly expanding, and heterogeneous evidence for cryotherapy as a minimally invasive alternative to surgery for breast cancer. The article provides a very clear and nuanced conclusion: cryotherapy appears safe and effective with encouraging local control and high patient satisfaction, but only for a very specific, "carefully selected" population (i.e., small, unifocal, hormone receptor-positive tumors).
The reviewer appreciates its methodological rigor that It adheres to PRISMA guidelines, was registered a priori in PROSPERO (a standard for high-quality systematic reviews), and involved independent screening and data extraction by two reviewers to minimize bias. The formal evidence appraisal is also worth mentioning that authors did not just summarize the literature; they critically appraised it. They used the MINORS tool to assess the risk of bias in the included studies and the GRADE framework to evaluate the overall certainty of the evidence for key outcomes.
The authors are transparent the primary shortcoming with the underlying body of evidence that it analyzed, stating that the "evidence base remains fragmented", including "marked variability" and "substantial heterogeneity" in how cryotherapy was performed across the 30 included studies. The authors also note that "follow-up duration was often <1 year", and "nearly half reported less than one year". This is not long enough to confidently assess long-term oncological outcomes like 5- or 10-year recurrence rates.
The reviewer also notes a few limitations and concerns, in no particular order:
•    Imaging accuracy after cryo needs
The abstract calls out imaging/reporting; in the body, post-cryo MRI/PET-CT are described and their financial burden noted, with a suggestion to consider CEM, but there’s no practical surveillance algorithm (modalities/timing/stop rules) or performance summary to serve as a convention.
In “3.3. Imaging Follow-Up and Response Assessment”, the manuscript notes imaging use, but clinicians need a concise statement on post-cryo false-negatives and the consequence for surveillance strategy (e.g., MRI limitations and fat-necrosis confounding). A table or paragraphs summarizing modality-specific pitfalls is recommended. 
Further in “4. Practical Implications (Clinician Take-aways)”, the review notes MRI/PET-CT roles and cost issues but doesn’t synthesize modality-specific accuracy or pitfalls (e.g., false-negatives after cryo, fat-necrosis confounders) into a surveillance algorithm.

•    Overlap/duplicate cohorts
In “3.1. Patient Selection and Tumor Characteristics”, multi-center series and national experiences sometimes re-report patients. I would recommend steps to detect/mitigate double-counting (even if only via author/institution/date cross-checks).

•    De-escalation comparison
In “4. Practical Implications (Clinician Take-aways)”, other non-surgical strategies (RT, endocrine therapy) were mentioned.  However, a high-level comparative lens (efficacy, toxicity, QoL, logistics) is not provided. A brief table positioning cryo vs standard breast-conserving surgery + RT vs endocrine-alone in the most relevant population would elevate clinical utility is recommended. 
The paper briefly mentions other de-escalation strategies (RT, endocrine therapy) and cost-effectiveness needs, but doesn’t position cryo against these options with even a schematic benefit/risk/QoL comparison for the ICE3-like population.
The abstract explicitly ties the review to surgical de-escalation (“…define where cryotherapy can appropriately de-escalate surgery”). The body shows feasibility in highly selected cases but never builds a decision framework or side-by-side comparator view (vs lumpectomy ± RT, endocrine alone) to make de-escalation actionable.

•    Device/protocol standardization
The introduction promises standardization needs; the discussion repeats that device/protocol heterogeneity is high (argon vs LN₂; cycles; margins) but stops short of proposing a concrete protocol or reporting template (e.g., required margin definition, probe count notation, freeze–thaw timing schema, mandatory follow-up windows).
In “3.2. Technique and Protocol for Cryotherapy”, the variations (argon vs LN₂, probe sizes, cycles, margin definitions 0.5-1.5 cm) were noted. The reviewer recommends proposing a provisional technical standard (e.g., double-freeze, target >=10 mm iceball beyond tumor, mandatory skin-safety measures) backed by the distribution reported, so that standardization claims would benefit from specifics. 
In another words, currently technique heterogeneity is acknowledged but under-quantified; the review calls for standardization but stops short of providing data-driven strata or a provisional technical standard readers could adopt. 

•    Mixing non-stratified results
In “3.5. Oncological Outcomes”, early-stage curative-intent cases are discussed together with metastatic/palliative settings, but outcomes aren’t consistently broken out by setting—this makes local control and OS numbers hard to interpret across very different intents of therapy.

•    Reporting completeness
A substantial fraction of studies did not report local recurrence. Complication summaries combine study-level statements with pooled percentages from a subset of studies, which could risk misinterpretation.

•    Health-economics
Cost burden of advanced imaging is flagged once and CEM is suggested as a cheaper option, but there’s no structured cost-effectiveness or budget-impact framing (index procedure + surveillance + retreatment).

•    Patient-reported outcomes (PROs)
The abstract says PROs are “needed”; the results acknowledge under-reporting in ~80% of studies and list “high satisfaction” in limited cohorts, but there is no synthesis of specific PRO instruments, time points, or core domains to standardize going forward.

Author Response

Comment 1: The abstract calls out imaging/reporting; in the body, post-cryo MRI/PET-CT are described and their financial burden noted, with a suggestion to consider CEM, but there’s no practical surveillance algorithm (modalities/timing/stop rules) or performance summary to serve as a convention.

Reply 1: Thank you for the pertinent comment. We have included this important point within our discussion and now further suggested a pragmatic framework that readers can consider at this point in time  Synthesizing these strengths and limitations suggests that post-cryoablation surveillance should be intentionally staged and risk-adapted. Early MRI (<3 months) risks false-negatives due to incomplete ablation zone maturation; therefore, a 3-month baseline MRI or CEM, followed by interval imaging at 9–18 months depending on BI-RADS category, may strike a balance between diagnostic accuracy and resource stewardship.  Escalation to advanced imaging or biopsy should be triggered by features inconsistent with expected cavity involution, such as new or enlarging enhancement, asymmetrical wall thickening, or metabolic activity not attributable to fat necrosis.

Comment 2: In “3.3. Imaging Follow-Up and Response Assessment”, the manuscript notes imaging use, but clinicians need a concise statement on post-cryo false-negatives and the consequence for surveillance strategy (e.g., MRI limitations and fat-necrosis confounding). A table or paragraphs summarizing modality-specific pitfalls is recommended. Further in “4. Practical Implications (Clinician Take-aways)”, the review notes MRI/PET-CT roles and cost issues but doesn’t synthesize modality-specific accuracy or pitfalls (e.g., false-negatives after cryo, fat-necrosis confounders) into a surveillance algorithm.

Reply 2: Thank you for the comment. We agree that clinicians benefit from practical guidance. We have added, “False-negative MRI examinations can occur, particularly when early scans are confounded by inflammatory changes or evolving fat necrosis, both of which may obscure subtle residual disease. Fat necrosis can also mimic either residual enhancement or obscure the ablation cavity’s expected involution pattern, making interpretation challenging even for experienced breast radiologists. PET-CT may offer additional value when metabolic activity is needed to differentiate viable tumor from post-treatment change, but it is expensive and generally less spatially precise for small-volume breast disease. The financial burden of serial high-cost imaging further complicates routine long-term monitoring for both patients and healthcare systems. This underscores the potential utility of cost-effective alternatives such as contrast-enhanced mammography (CEM), which may offer sufficient sensitivity for detecting suspicious enhancement while reducing cumulative imaging costs [47]. However, no study has yet provided a comprehensive cost-effectiveness evaluation comparing MRI, PET-CT, CEM, and multimodal strategies, representing a key gap in current evidence.”

Comment 3: In “3.1. Patient Selection and Tumor Characteristics”, multi-center series and national experiences sometimes re-report patients. I would recommend steps to detect/mitigate double-counting (even if only via author/institution/date cross-checks).

Reply 3: To minimise potential double-counting, we cross-checked study authorship, centres, enrolment periods, and cryotherapy devices; no clear duplicate cohorts were identified, but multi-centre registry-based studies were indicated in the results table (Table 1) where overlap could not be definitively excluded.

Comment 4: In “4. Practical Implications (Clinician Take-aways)”, other non-surgical strategies (RT, endocrine therapy) were mentioned. However, a high-level comparative lens (efficacy, toxicity, QoL, logistics) is not provided. A brief table positioning cryo vs standard breast-conserving surgery + RT vs endocrine-alone in the most relevant population would elevate clinical utility is recommended.

Reply 4: Despite the advantages of cryotherapy such as good cosmesis, minimal downtime, and outpatient feasibility, it should be viewed as complementary to, rather than a replacement for, established standards of care. Breast-conserving surgery with radiotherapy remains the primary modality for durable long-term local control, while endocrine therapy alone may be appropriate only for highly selected low-risk luminal A patients. Table 3 outlines a comparative overview of these de-escalation approaches to guide clinical decision-making in populations similar to those in ICE3.

Comment 5: The paper briefly mentions other de-escalation strategies (RT, endocrine therapy) and cost-effectiveness needs, but doesn’t position cryo against these options with even a schematic benefit/risk/QoL comparison for the ICE3-like population.

Reply 5: We thank the reviewer for this insightful suggestion. In response, we have now added a new comparative table (“Table 3. Comparative Overview of De-escalation Strategies for Low-Risk Early Breast Cancer”) that directly contrasts cryotherapy, breast-conserving surgery plus radiotherapy, and endocrine-therapy-alone regimens often considered for highly selected luminal A patients.

Comment 6: The abstract explicitly ties the review to surgical de-escalation (“…define where cryotherapy can appropriately de-escalate surgery”). The body shows feasibility in highly selected cases but never builds a decision framework or side-by-side comparator view (vs lumpectomy ± RT, endocrine alone) to make de-escalation actionable.

Reply 6: We have now added a dedicated subsection synthesizing how cryotherapy aligns with, complements, or diverges from established options such as lumpectomy with radiotherapy and endocrine therapy alone.

Comment 7: The introduction promises standardization needs; the discussion repeats that device/protocol heterogeneity is high (argon vs LN₂; cycles; margins) but stops short of proposing a concrete protocol or reporting template (e.g., required margin definition, probe count notation, freeze–thaw timing schema, mandatory follow-up windows). In “3.2. Technique and Protocol for Cryotherapy”, the variations (argon vs LN₂, probe sizes, cycles, margin definitions 0.5-1.5 cm) were noted. The reviewer recommends proposing a provisional technical standard (e.g., double-freeze, target >=10 mm iceball beyond tumor, mandatory skin-safety measures) backed by the distribution reported, so that standardization claims would benefit from specifics. In another words, currently technique heterogeneity is acknowledged but under-quantified; the review calls for standardization but stops short of providing data-driven strata or a provisional technical standard readers could adopt.

Reply 7: To address these inconsistencies, we propose a provisional technical framework derived from the most frequently reported parameters across included studies. At minimum, protocols should incorporate: (1) a double freeze–thaw cycle; (2) a target iceball margin ≥10 mm beyond the tumor; (3) explicit documentation of probe count and configuration, with guidance for multi-probe placement in larger tumors; (4) standardized reporting of freeze and thaw timing, whether fixed or margin-driven; and (5) mandatory skin-protection measures, with warmed saline bags recommended as the preferred technique. We further suggest a reporting template that includes detailed documentation of tumor size, distance to skin, intended margins, probe type and number, device system, real-time monitoring modality, and standardized follow-up imaging windows

Comment 8: In “3.5. Oncological Outcomes”, early-stage curative-intent cases are discussed together with metastatic/palliative settings, but outcomes aren’t consistently broken out by setting—this makes local control and OS numbers hard to interpret across very different intents of therapy.

Reply 8: Thank you for the comment. We have now further explained that, "Follow-up duration across studies ranged from 1 day [32,33] to 10 years [18]. Of the included cohorts, 13 studies (43%) had <1 year follow-up, 7 (23%) had <2 years, and 4 (13%) had <3 years, limiting interpretation of long-term oncologic outcomes. Among the 28/30 studies evaluating cryotherapy for primary, early-stage breast cancer, local recurrence rates varied widely: 9 studies (32%) reported 0% recurrence, 5 (18%) reported <10%, 1 (4%) reported <20%, 3 (11%) reported <30%, and 1 (4%) reported <40%. Notably, 9 studies (32%) did not report local recurrence at all. Six early-stage studies reported managing recurrence with repeat cryotherapy [19,25,30,37–39]. Distant recurrence in this curative-intent group was reported in 14 studies, with 11 showing 0%, 2 reporting <10%, and 1 reporting <20%. Vogl et al. [40] reported the highest rate of intramammary distant recurrence (13%, n = 6)."

Overall survival (OS) was inconsistently reported: only 13/28 (46%) early-stage studies documented OS outcomes, with 9 studies reporting 100% OS and four reporting OS between 80–92% [25,35,40]. These figures must be interpreted cautiously given short follow-up and highly selected low-risk populations.

In contrast, outcomes from metastatic or palliative-intent cohorts were reported separately and cannot be interpreted alongside curative-intent data. Liang et al. [21], examining HER2-overexpressing recurrent disease, demonstrated longer progression-free survival in the combination cryotherapy–NK cell therapy group compared with cryotherapy alone (12 vs. 9 months), consistent with treatment administered for disease control rather than definitive local cure.

Comment 9: A substantial fraction of studies did not report local recurrence. Complication summaries combine study-level statements with pooled percentages from a subset of studies, which could risk misinterpretation.

Reply 9: We agree and have now provided explicit denominators for every recurrence and complication, ensuring that all percentages are clearly tied to the number of studies that reported the outcome; and also declared the number of studies that did not report complications.

We also added a cautionary statement in the Discussion acknowledging the substantial number of studies with missing d’ata and the resulting limitations in interpreting pooled rates or comparing outcomes across cohorts. In terms of complications, hematomas and bruising were the most common, though up to 80% of patients in some studies reported no adverse events, raising questions about reporting biases and the importance of rigorous follow-up to capture adverse events comprehensively [16,18,29,37].”

Comment 10: Cost burden of advanced imaging is flagged once and CEM is suggested as a cheaper option, but there’s no structured cost-effectiveness or budget-impact framing (index procedure + surveillance + retreatment).

Reply 10: Thank you for the important observation. We have further elaborated in the Discussion that, “The financial burden of serial high-cost imaging further complicates routine long-term monitoring for both patients and healthcare systems. This underscores the potential utility of cost-effective alternatives such as contrast-enhanced mammography (CEM), which may offer sufficient sensitivity for detecting suspicious enhancement while reducing cumulative imaging costs [47]. However, no study has yet provided a comprehensive cost-effectiveness evaluation comparing MRI, PET-CT, CEM, and multimodal strategies, representing a key gap in current evidence.”

Comment 11: The abstract says PROs are “needed”; the results acknowledge under-reporting in ~80% of studies and list “high satisfaction” in limited cohorts, but there is no synthesis of specific PRO instruments, time points, or core domains to standardize going forward.

Reply 11: Thank you for the comment. We have now further elaborated in the Discussion that, “Patient-reported outcomes were reported in only six studies, and the use of heterogeneous instruments (EQ-5D-5L, EuroQol VAS, 5-point cosmesis scales, KPS) contributed to variability in outcome measurement. Although commonly evaluated domains included pain, appearance, and functional recovery, the field would benefit from the development and adoption of a standardized core patient-reported outcomes set to ensure consistency and comparability across future trials.”

Round 2

Reviewer 2 Report

Comments and Suggestions for Authors

The authors have been highly responsive. They did not just make minor text edits; they generated new tables (Table 3), proposed new frameworks (surveillance algorithm, technical protocol), and restructured their data analysis to be more rigorous. The manuscript "v2" is a substantially improved document that directly implements the peer review feedback.